# Multi-Domain Graph Foundation Models: Robust Knowledge Transfer via Topology Alignment

**Shuo Wang** [*1] **Bokui Wang** [*1] **Zhixiang Shen** [*1] **Boyan Deng** [1] **Zhao Kang** [†1]

## Abstract

Recent advances in CV and NLP have inspired researchers to develop general-purpose graph foundation models through pre-training across diverse domains. However, a fundamental challenge arises from the substantial differences in graph topologies across domains. Additionally, real-world graphs are often sparse and prone to noisy connections and adversarial attacks. To address these issues, we propose the Multi-Domain Graph Foundation Model (MDGFM), a unified framework that aligns and leverages cross-domain topological information to facilitate effective and robust knowledge transfer. MDGFM bridges different domains by adaptively balancing features and topology while refining original graphs to eliminate noise and align topological structures. To further enhance knowledge transfer, we introduce an efficient prompt-tuning approach. By aligning topologies, MDGFM not only improves multi-domain pre-training but also enables robust knowledge transfer to unseen domains. Theoretical analyses provide guarantees of MDGFM's effectiveness and domain generalization capabilities. Extensive experiments on both homophilic and heterophilic graph datasets validate the robustness and efficacy of our method. Our code is available at https://github.com/wbkzwqtzw/MDGFM.

## 1. Introduction

Graphs, as a versatile data structure, are widely used across various domains, such as citation networks (Ebesu & Fang, 2017), social networks (Traud et al., 2012), and bioinformat-

ics (Zhang et al., 2021). Inspired by recent advances in CV and NLP (Vidit et al., 2023; Cheng et al., 2024), researchers have sought to develop general-purpose graph foundation models. In particular, multi-domain graph pre-training has gained significant attention for its ability to integrate knowledge from various domains and enable effective transfer learning (Yu et al., 2025; Zhang et al., 2024b). This approach is viewed as a critical milestone toward the creation of truly general-purpose graph models.

Despite significant progress, a substantial gap remains in fully understanding the richness and diversity of graph structural knowledge. Existing methods primarily rely on fixed graph topologies and apply uniform encoding mechanisms across all domains (Zhao et al., 2024; Yu et al., 2025), which severely limits their generalizability across diverse domains (Zhang et al., 2024). In this paper, we revisit the multi-domain graph foundation model from a structural perspective and address a critical challenge: topology alignment across different domains. Achieving effective topology alignment is a non-trivial task, presenting two primary challenges that must be overcome to advance toward a truly general-purpose graph model.

First, structural knowledge across domains often exhibits significant semantic differences. For instance, citation networks predominantly display highly homophilic patterns, whereas social networks and webpage link graphs frequently contain numerous heterophilic edges (Zheng et al., 2022; Shen and Kang, 2025; Xie et al., 2025). These differences necessitate domain-specific adaptations of encoding mechanisms (Li et al., 2024; Pan and Kang, 2023). As a result, there is an urgent need for a unified framework that can adaptively capture critical information from both features and topology while effectively learning domain-invariant knowledge. Such a framework is essential to ensure robust generalization to downstream domains (Li et al., 2022; Xie et al., 2025).

Second, real-world graphs are inherently noisy (Jin et al., 2020b), often containing unreliable edges characterized by irrelevant, misleading, or missing connections. Additionally, graph learning algorithms are highly susceptible to adversarial attacks (Zügner et al., 2020), further exacerbating the challenges of graph representation learning. These issues

---

[*]Equal contribution [†]Corresponding author [1]University of Electronic Science and Technology of China, Chengdu, Sichuan Province, China. Correspondence to: Shuo Wang <runner21st@gmail.com>, Zhao Kang <zkang@uestc.edu.cn>.

*Proceedings of the 42nd International Conference on Machine Learning*, Vancouver, Canada. PMLR 267, 2025. Copyright 2025 by the author(s).

highlight the limitations of conventional training paradigms that rely on fixed graph topologies. Therefore, developing a robust multi-domain pre-training framework capable of effectively handling complex noise distributions is critical for constructing trustworthy and reliable graph foundation models.

To tackle these challenges, we propose the Multi-Domain Graph Foundation Model (MDGFM), a unified framework designed to effectively align and leverage structural knowledge across domains. First, we introduce a decoupled embedding mechanism incorporating an adaptive balance token that dynamically weighs feature and topology information. To address inherent noise and achieve topology alignment, we integrate a Graph Structure Learning module that learns robust, domain-invariant knowledge, enhancing both generalization and transferability. Additionally, we develop an efficient prompt learning strategy to transfer knowledge from multiple source domains to a target domain. By aligning topologies effectively, MDGFM not only improves multi-domain pre-training but also enables efficient and robust knowledge transfer to unseen downstream domains. Theoretical analyses validate the effectiveness of our topological knowledge alignment and demonstrate superior domain generalization capabilities.

In summary, our contributions are as follows:

- We introduce a novel structural perspective on multi-domain graph foundation models, addressing the complexities arising from interconnected nodes and diverse attribute types.

- We propose MDGFM, a unified multi-domain graph foundation model that employs graph structure learning to reduce inherent noise and align topological structures. Our framework facilitates efficient and robust knowledge transfer to unseen domains, with theoretical guarantees on its effectiveness and domain generalization capabilities.

- We conduct extensive experiments on both homophilic and heterophilic benchmark graph datasets, demonstrating the effectiveness, robustness, and versatility of our approach.

## 2. Related work

### 2.1. Graph Pre-training

Graph pre-training aims to train Graph Neural Networks (GNNs) on large amounts of unlabeled data, enabling the transfer of the learned model to downstream tasks with limited supervision. This approach facilitates the acquisition of general knowledge about real-world graphs while reducing the dependence on labeled data (Hu et al., 2020; Jin et al.,

2020a).

Pre-training methods can be broadly categorized based on their downstream task-tuning strategies. The first category follows a pretrain-and-fine-tune paradigm, emphasizing effective pre-training strategies in the upstream phase through self-supervised learning. For example, DGI (Velickovic et al., 2019) enhances training by maximizing the mutual information between global and local representations. Similarly, GraphCL (You et al., 2020) and SimGRACE (Xia et al., 2022) focus on minimizing the distance between representations of different augmentations, effectively capturing invariant and robust structural information.

The second type follows a pretrain-and-prompt-tuning paradigm, where pre-trained models are not fine-tuned for downstream tasks. Instead, these methods reformulate the input data to align with the pretext task (Gao et al., 2020). For instance, GPPT (Sun et al., 2022a) introduces a graph prompting function that transforms independent nodes into token pairs, reframing downstream node classification as an edge prediction task. Similarly, GPF (Fang et al., 2024) employs learnable perturbations in the feature space of downstream graphs, enabling implicit modifications to node features and graph structures. However, most existing methods are constrained to single-domain pre-training and tuning, significantly limiting their capacity to capture cross-domain knowledge and generalize to unseen domains.

### 2.2. Multi-domain Generalization

Domain generalization aims to achieve out-of-distribution (OOD) generalization by learning from multiple source domains (Zhou et al., 2022). In contrast to domain adaptation—which transfers prior knowledge from a single source domain to a specific target domain—domain generalization focuses on leveraging diverse information from multiple source domains to generalize effectively to unseen domains. This approach addresses two critical challenges: domain shift and the absence of target domain data (Blanchard et al., 2011).

Recently, domain generalization on graphs has gained attention (Wang et al., 2024). These methods integrate and extract knowledge from multiple source domains during the upstream pre-training phase, enabling the transfer of this knowledge to tackle various graph-related tasks in previously unseen downstream domains (Fang et al., 2025). For example, GCOPE (Zhao et al., 2024) integrates multi-source graph topologies during the pre-training phase by introducing interconnected virtual nodes. Additionally, MDGPT (Yu et al., 2025) incorporates domain-specific tokens in the pre-training phase to align node features from different domains, and employs prompt-tuning during downstream tasks for efficient knowledge transfer. Despite these advancements, significant gaps remain in understanding the

semantic differences and reliability of multi-domain topologies. Consequently, there is an urgent need to design a universal foundation model capable of achieving robust and generalized knowledge transfer across domains.

## 3. Problem Definition

A graph is formally represented as $G = (V, E, X^{ori}) = (A, X^{ori})$, where $V$ is the set of nodes and $E$ denotes the set of edges, $X^{ori} \in \mathbb{R}^{|V| \times d'}$ represents the original feature matrix of nodes. $A$ is the corresponding adjacency matrix. Each graph is associated with a domain $D_i \in \mathcal{D}$. Without loss of generality, we assume $D_i \neq D_j$ for any pair of distinct graphs.

Given a set of graphs from different domains, say $G_i = (A_i, X_i^{ori}), i = 1, ..., N$. We pretrain our graph foundation model on all visible graphs and test the performance on graph $T$ from an unseen target domain $D_T \notin \mathcal{D}$. We summarize the notations in Appendix A.

## 4. Methodology

As illustrated in Figure 1, our proposed MDGFM consists of two modules: the *Multi-domain pre-training* module and the *Downstream target-domain adaptation* module.

**Multi-domain pre-training.** For graphs originating from multiple source domains, we begin by unifying their feature matrices to a common dimensionality, ensuring compatibility across domains. Next, we apply token operators specifically designed to achieve semantic alignment between disparate graphs. To address the challenges posed by heterophily in downstream tasks, we leverage graph structure learning (GSL) to refine each source domain, integrating both feature and topology information during the refinement process. Additionally, GSL enables the model to learn domain-invariant knowledge for better transfer, where we prove the effectiveness in Section 5. Finally, the graph encoder is trained effectively using self-supervised signals, enabling robust representation learning.

**Downstream target-domain adaptation.** Given that graphs often share common intrinsic patterns, pre-trained knowledge can be effectively transferred to unseen graphs (Zhao et al., 2024). To bridge the gap between source and target domains, we propose a dual-stream prompt framework. This framework combines meta prompts, which transfer broadly learned knowledge, with task-specific prompts designed to align with the unique characteristics of the downstream domain.

### 4.1. Feature Projection into Unified Semantic Space

Graphs from different domains often exhibit diverse features and structural patterns. To address these discrepan-

cies and customize the information propagation process in GNNs, we employ both numerical and semantic alignment to bridge domain gaps. First, we transform the feature matrix of each graph into a consistent dimensionality shared across all domains, enabling uniform representation. This transformation is described by:

$$X_i = Proj(X_i^{ori}) \in \mathbb{R}^{|V_i| \times d} \tag{1}$$

where $d$ represents the aligned dimensionality, and $Proj(\cdot)$ denotes a specific projection operation. In our approach, the *Projection* function is implemented using Principal Component Analysis (PCA) (Abdi & Williams, 2010). While this dimensional alignment ensures uniform feature dimensions across graphs, it primarily addresses numerical consistency and does not resolve the semantic disparities that remain between domains.

To achieve semantic alignment, we propose the concept of *domain tokens* denoted as $t_{D_i} \in \mathbb{R}^d$, which encode domain-specific characteristics by capturing unique contextual information from each domain. These tokens are applied via element-wise Hadamard multiplication $\odot$ with the feature matrices, acting as adaptive filters that modulate features to reflect their respective domain properties within a unified semantic space. This process not only preserves the distinctive traits of individual domains but also facilitates their alignment within a cohesive semantic framework.

A significant challenge in building a graph foundation model lies in the substantial semantic variations across domains, which can hinder effective knowledge transfer (Hassani, 2022). To address this, we leverage shared, domain-agnostic information as a critical foundation for robust and efficient transfer learning. Specifically, we introduce the *shared token* $t_S \in \mathbb{R}^d$ which acts as a common semantic anchor to bridge discrepancies between domains. By focusing on shared patterns, $t_s$ captures transferable knowledge broadly applicable to new domains, reducing dependence on domain-specific features that may lack generalizability. This shared representation enables consistent semantic alignment, enhancing the model's adaptability and robustness to domain variations. The overall feature unification procedure can be formalized as:

$$X_i' = t_S \odot \sigma(t_{D_i} \odot X_i) \tag{2}$$

where $\sigma(\cdot)$ is non-linearity activation function to capture complex information.

### 4.2. Graph Topology-aware Alignment

Previous research on graph prompting (Sun et al., 2023; Dong et al., 2019) has predominantly emphasized feature alignment, often overlooking the critical role of structural discrepancies between graphs from different domains. However, rich semantic information is embedded within relational patterns and topology credibility, as characterized by

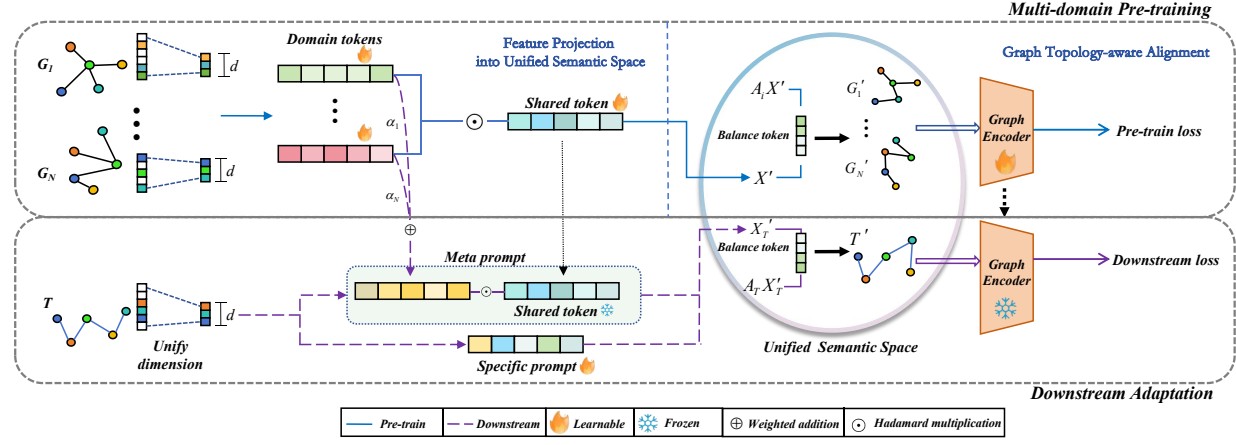

*Figure 1.* The overall framework of the proposed MDGFM.

metrics like homophily and heterophily edge ratios (Zhu et al., 2020). These structural characteristics underscore the importance of aligning graph structure patterns to enable meaningful knowledge transfer (Sun et al., 2022b; Zheng et al., 2022).

Rather than solely focusing on direct structural unification, our approach seeks to synchronize more reliable topology information through graph structure refinement on both source and unseen downstream graphs. This is achieved via the incorporation of GSL, which enhances the robustness and effectiveness of cross-domain graph alignment. By addressing structural discrepancies, our framework bridges the gap between diverse graph domains, facilitating more comprehensive knowledge transfer.

Unlike prior GSL approaches (Zhu et al., 2021; Jin et al., 2020b), which primarily optimize graph structures based on feature similarity, our method incorporates both semantic and topological information for a more comprehensive refinement process. Specifically, we utilize $A_i^r X_i'$ to capture the original structural information, where $r$ represents the order of graph aggregation (default set to one). To effectively fuse $X_i'$ and $A_i^r X_i'$ during graph refinement, we introduce a *balance token* $t_{B_i} \in \mathbb{R}^{2d}$, which operates in the following manner:

$$H_i = t_{B_i} \odot [X_i', A_i^r X_i'] \qquad (3)$$

Here, $t_{B_i}$ dynamically balances the contributions of node features and aggregated structural information.

Finally, consistent with existing GSL methods (Li et al., 2024), we apply post-processing techniques to reconstruct the refined adjacency matrix. Specifically, a similarity matrix is first computed using $H_i$ and then sparsified via $k$-nearest neighbors ($k$NN). In practice, we employ $k$NN sparsification with its locality-sensitive approximation to enhance efficiency (Fatemi et al., 2021). Subsequently, operations such as Symmetrization, Activation, and Normal-

ization are performed sequentially to produce the final $A_i'$. Detailed implementation steps are provided in Appendix F. Building on this topology-aware refinement process, we propose a general pretraining framework capable of generating high-quality embeddings across multiple domains using contrastive learning. By optimizing a contrastive-based objective, the model learns generalized representations by leveraging rich sample-to-sample relationships from diverse perspectives (Yao et al., 2022). Mathematically, we maximize the mutual information between $G_{i1} = (A_i, X_i')$ and $G_{i2} = (A_i', X_i')$ as follows:

$$\mathcal{L} = -I(G_{i1}; G_{i2} \dagger I_e) - I(G_{i1}; G_{i2} \dagger A_i') \qquad (4)$$

where we use $\dagger$ to indicate positive samples when computing similarities, often utilizing the identity matrix $I_e$ for this purpose. The second term of $\mathcal{L}$ incorporates the refined graph structure $A_i'$ to increase the number of positive samples. Detailed computations of the loss function are provided in Appendix G. This loss formulation reduces the information gap between the original and refined graphs, ensuring that global structural properties are preserved during optimization. By maintaining semantic consistency in the graph structure, it enhances alignment and adaptability for downstream tasks.

### 4.3. Knowledge Transfer to Downstream Domain

Similar to in-context learning, downstream adaptation aims to enhance a model's ability to learn tasks using only a few examples provided as demonstrations (Dong et al., 2024), highlighting the need for upstream knowledge transfer. Inspired by the "pre-training & prompting" paradigm, our framework employs a dual-prompt strategy. Specifically, the **meta prompt** $p_m$ focuses on adjusting the distribution of learned knowledge by modeling the relationships between the target and source domains. This can be formalized as a

function defined for feature $X$:

$$p_m(X) = t_S \odot \sigma(\sum_{i=1}^{N} \alpha_i t_{D_i} \odot X) \qquad (5)$$

where $\alpha_1, ..., \alpha_N$ are trainable coefficients for the source domains. Meanwhile, the learnable **specific prompt** $p_s \in \mathbb{R}^d$ aims to directly align the target domain with the unified semantic space by learning from the limited available samples, ensuring precise adaptation to the downstream domain.

As discussed, for the downstream graph $T(A_T, X_T)$ where the feature dimension aligns with the given $d$, the dual prompts $p_m$ and $p_s$ operate on $X_T$ to achieve semantic unification. Since all tokens are optimized during the pretraining phase, only the coefficients and $p_s$ need to be trained in this process. The reduced number of learnable parameters enhances transfer performance by compensating for the scarcity of annotated samples. Consistent with upstream operations, we use $H_T = t \odot [X'_T, A^r_T X'_T]$ to obtain $A'_T$, where $t$ could be composed of $t_{D_i}$ (similar to $p_m$) or be directly trained. Finally, we obtain the node representations $Z$ as follows, which can be applied to downstream tasks:

$$Z = GE(A'_T, \beta p_m(X_T) + (1-\beta)p_s \odot X_T; \theta_{pre}) \quad (6)$$

where $GE$ denotes the graph encoder and $\theta_{pre}$ the frozen parameters learned in pre-training phase. To mitigate the risk of harmful transfer from source graphs with significant discrepancies, as discussed by (Yan et al., 2024), the prompts are integrated using a learnable parameter, $\beta$, instead of relying solely on concatenation. For graphs that significantly differ from the source domains, the meta prompt weights are attenuated to minimize the transfer of irrelevant information, thereby reducing its potential negative impact on downstream tasks.

For the downstream node classification task, the loss function $\mathcal{L}_{dst}$ follows a universal task template grounded in subgraph similarity. Given a labeled training set $S = \{(x_1, y_1), \ldots, (x_i, y_i), \ldots\}$, where $x_i$ represents a node and $y_i \in Y$ denotes the class label of $x_i$ with $Y$ denoting the set of all possible class labels, the loss function is expressed as:

$$\mathcal{L}_{dst} = -\sum_{(x_i, y_i) \in S} \ln \frac{\frac{1}{\tau} sim(z_{x_i}, \bar{z}_{y_i})}{\sum_{y \in Y} \exp\left(\frac{1}{\tau} sim(z_{x_i}, \bar{z}_y)\right)} \quad (7)$$

Here, $z_{x_i}$ denotes the final embedding of node $x_i$, and $\bar{z}_y$ represents the class embedding for class $y$, computed as the mean embedding of all training instances belonging to $y$. We practically calculate $sim(\cdot, \cdot)$ by cosine similarity, while $\tau$ serves as a temperature parameter to control the shape of the output distribution.

## 5. Theoretical Analysis

In this section, we provide a theoretical analysis to demonstrate the domain generalization capabilities of MDGFM.

Denote $P^t_\mathcal{X}$ as the data distribution on feature space in the target domain. $t$ could be changed into $i$ for source domain, while $\mathcal{X}$ could be replaced by label space $\mathcal{Y}$. With the covariate shift assumption, each domain is characterized by the distribution on $\mathcal{X}$. Thus, we can approximate the target domain distribution $P^t_\mathcal{X}$ within the convex hull of source domain distributions: $\Lambda := \{\sum_{i=1}^{M} \pi_i P^i_\mathcal{X} \mid \pi \in \Delta_M\}$, where $\Delta_M$ is the $(M-1)$-dimensional simplex so that each $\pi$ represents a normalized mixing weights. The generalization capability is quantified by the following:

**Theorem 5.1** (Domain generalization error bound). *Let $\gamma := \min_{\pi \in \Delta_M} d_\mathcal{H}(P^t_\mathcal{X}, \sum_{i=1}^{M} \pi_i P^i_\mathcal{X})$ with minimizer $\pi^*$ be the distance of $P^t_\mathcal{X}$ from the convex hull $\Lambda$, and $P^*_\mathcal{X} := \sum_{i=1}^{M} \pi^*_i P^i_\mathcal{X}$ be the best approximator within $\Lambda$. Let $\rho := \sup_{P'_\mathcal{X}, P''_\mathcal{X} \in \Lambda} d_\mathcal{H}(P'_\mathcal{X}, P''_\mathcal{X})$ be the diameter of $\Lambda$. Then it holds that*

$$\epsilon^t(h) \leq \sum_{i=1}^{M} \pi^*_i \epsilon^i(h) + \frac{\gamma + \rho}{2} + \lambda_{\mathcal{H}, (P^t_\mathcal{X}, P^*_\mathcal{X})}, \quad (8)$$

*where $\lambda_{\mathcal{H}, (P^t_\mathcal{X}, P^*_\mathcal{X})}$ is the ideal joint risk across the target domain and the domain with the best approximator distribution $P^*_\mathcal{X}$. $\epsilon^1, \cdots, \epsilon^M$ represent the source risks and $\epsilon^t$ denotes the target risk (Albuquerque et al., 2019).*

To minimize the upper error bound, we introduce the concept of invariant graph learning, which is highlighted in our work:

**Assumption 5.2.** Given a graph $G$, there exists an optimal invariant graph learner $\Phi^*(G)$ satisfying:
*Invariance Property:* $\forall e, P^e(Y|\Phi^*(G)) = P^{e'}(Y|\Phi^*(G))$.
*Sufficient Property:* $Y = \omega^*(g^*(\Phi^*(G))) + \epsilon, \epsilon \perp G$.
where $g^*(\cdot)$ is a representation learning function, $\omega^*$ is the classifier, $e$ denotes the environments (i.e., domains), $\perp$ indicates statistical independence and $\epsilon$ is the random noise.

$\Phi^*(G)$ could generate invariant graphs across different domains, which is implemented as GSL procedure on all domains in our work. Moreover, by maximizing the mutual information between $G_{i1}(A_i, X'_i)$ and $G_{i2}(A'_i, X'_i)$, our method retains sufficient task-relevant information, thereby satisfying the sufficient property. Motivated by Theorem 5.1, domain invariant representation learning minimizes the risks over all source domains corresponding to the first term of the bound, as well as the representation distribution differences among source and target domains in the hope of reducing $\gamma$ and $\rho$. In general, based on the following theorem, our model achieves the **minimal error bound** due to its adherence to the invariant and sufficient properties.

**Theorem 5.3.** *Let $\Phi^*$ be the optimal invariant graph learner and denote the complement as $G\backslash\Phi^*(G)$, i.e., the corresponding variant subgraph. Then we can obtain the optimal predictor under distribution shifts as follows.*

$$\arg\min_{\omega,g} \omega \circ g \circ \Phi^*(G) = \arg\min_{f} \sup_{e} \mathcal{R}(f|e) \quad (9)$$

*if the following conditions hold: (1) $\Phi^*(G) \perp G\backslash\Phi^*(G)$; (2) $\forall\Phi$ which is an invariant graph learner, $\exists e'$ such that $P^{e'}(G,Y) = P^{e'}(\Phi(G),Y)P^{e'}(G\backslash\Phi(G))$ and $P^{e'}(\Phi(G)) = P^e(\Phi(G))$ (Li et al., 2022).*

where $\mathcal{R}(f|e) = \mathbb{E}_{G,Y}^e[l(f(G),Y)]$ is the risk of the predictor $f$ on the domain $e$ and $l(\cdot,\cdot) : \mathbb{Y} \times \mathbb{Y} \to \mathbb{R}$ denotes a loss function.

# 6. Experiments

In this section, we evaluate our proposed MDGFM on few-shot node classification task. Specifically, we aim to answer the following four research questions:

**RQ1.** How does MDGFM perform on multi-graph tasks in few-shot learning scenarios compared to current baselines? **RQ2.** How do the key components benefit our model? **RQ3.** Does our model rely on specific source domains and exhibit sensitivity to the removal of these source domains? **RQ4.** Does our model demonstrate robustness against attacks and deletion on the source domain?

## 6.1. Experimental Setups

**Datasets.** To ensure a comprehensive comparison, we conduct experiments on six primary datasets, including three homophilic graphs—Cora (Sen et al., 2008), Citeseer (Sen et al., 2008), and Pubmed (Namata et al., 2012)—and three heterophilic graphs—Cornell, Chameleon, and Squirrel (Pei et al., 2020). Additionally, we include Penn94 (Traud et al., 2012), a large-scale graph dataset, as a downstream target domain. Detailed statistical information on these datasets is provided in Appendix C.

**Baselines**. We compare our method against four categories of approaches:
**Supervised methods**: These methods train a GNN on downstream tasks and directly infer results. We employ two well-known models: GCN (Kipf & Welling, 2016) and GAT (Velickovic et al., 2017).
**Graph Pre-training Methods**: These approaches perform self-supervised pre-training across multiple isolated source domains, such as DGI (Velickovic et al., 2019) and GraphCL (You et al., 2020), before fine-tuning on a new downstream task. Notably, source domains are not trained simultaneously; instead, they are merged into a single batch object, forming an adjacency matrix composed of distinct blocks.
**Graph Prompting Methods**: These methods freeze the parameters of a pre-trained model, unify downstream tasks and tune a single type of prompt accordingly. Representative methods include GPPT (Sun et al., 2022a) and GPF (Fang et al., 2024).
**Multi-domain Pre-training Methods**: These methods integrate multiple source domains during upstream pre-training and transfer knowledge for few-shot learning on unseen target domains. For instance, GCOPE (Zhao et al., 2024) constructs a unified large-scale dataset with inter-dataset connections, while MDGPT (Yu et al., 2025) encodes domain-specific characteristics using unique tokens, followed by fine-tuning and prompt adaptation.

For a fair comparison, we use GCN as the backbone for all methods. To evaluate performance on unseen target domains, we focus on scenarios where the model generalizes to domains not encountered during pre-training. For the Cora, Citeseer, Pubmed, Chameleon, Squirrel, and Cornell datasets, we designate one dataset as the downstream target domain while using the remaining five as source domains during pre-training. Additionally, we extend this setup by using all six datasets as source domains and applying Penn94 as the target domain for few-shot learning. More details are provided in Appendix E.

## 6.2. Cross-domain Transfer Efficacy under Few-shot Learning Conditions (RQ1)

We compare our proposed MDGFM against all baselines on 1-shot and $K$-shot node classification tasks. Each experiment is repeated five times, and we report the average results in Tables 1 and 2.

In the one-shot setting, MDGFM consistently outperforms all baseline models, demonstrating superior performance on both homophilic and heterophilic graphs. This advantage stems from MDGFM's ability to effectively capture both domain-specific information from each source domain and shared patterns across domains. Notably, we observe that supervised methods occasionally surpass graph pre-training approaches, suggesting the presence of negative transfer in certain scenarios.

Among multi-domain pre-training methods, MDGPT outperforms GCOPE on homophilic graphs, as GCOPE's reliance on virtual nodes propagates homophilic graph information, inadvertently introducing noise. Conversely, on heterophilic graphs, GCOPE surpasses MDGPT, as MDGPT's simple integration of source domain tokens fails to fully capture cross-domain commonalities. For graph prompting methods, GPPT generally underperforms across most datasets, as it is not specifically designed for few-shot learning. While GPF achieves competitive performance, its high memory requirements pose a significant limitation.

In the few-shot setting, we observe trends similar to those

*Table 1.* Cross-domain transfer learning performance (mean accuracy $\pm$ std) of one-shot node classification. The highest result is **bolded** and the runner-up is highlighted with red color. The symbol "OOM" means out of memory.

| Methods | Cora | Citeseer | Pubmed | Cornell | Squirrel | Chameleon | Penn94 |
|---|---|---|---|---|---|---|---|
| GCN | 28.57±5.07 | 31.27±4.53 | 40.55±5.65 | 31.81±4.71 | 20.00±0.29 | 24.17±5.21 | 50.45±1.79 |
| GAT | 28.40±6.25 | 30.76±5.40 | 39.99±4.96 | 28.03±13.19 | 21.55±2.30 | 23.93±4.11 | 50.48±1.40 |
| DGI | 29.30±5.82 | 30.03±4.88 | 41.85±7.78 | 31.54±15.66 | 21.15±1.68 | 21.73±5.47 | 50.22±0.74 |
| GraphCL | 34.94±6.49 | 30.58±4.58 | 40.37±7.81 | 27.15±12.64 | 21.42±2.23 | 22.49±3.02 | 50.61±2.03 |
| GPPT | 17.52±5.52 | 21.45±3.45 | 36.56±5.31 | 25.09±2.92 | 20.09±0.91 | 24.53±2.55 | 48.93±1.39 |
| GPF | 37.84±11.07 | 37.61±8.87 | 46.36±7.48 | 34.54±7.73 | 21.92±3.50 | 25.90±8.51 | OOM |
| GCOPE | 34.23±8.16 | 39.05±8.82 | 44.85±6.72 | 34.02±11.94 | 22.46±1.96 | 24.61±3.99 | 50.79±0.65 |
| MDGPT | 39.54±9.02 | 39.24±8.95 | 45.39±11.01 | 33.58±10.38 | 22.35±3.77 | 23.68±1.56 | 50.78±3.05 |
| MDGFM | **44.83±7.41** | **42.18±6.41** | **46.84±7.31** | **40.77±5.96** | **24.30±3.26** | **28.36±3.65** | **52.36±0.86** |

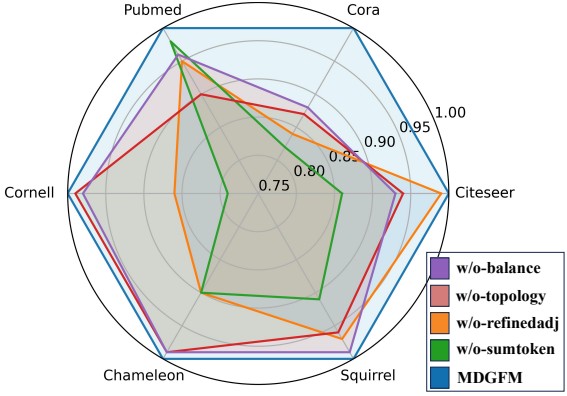

*Figure 2.* Ablation studies on key components.

in the one-shot scenario. Notably, on heterophilic graphs, certain methods exhibit a decline in performance compared to the one-shot case. This degradation is primarily due to the introduction of noise from the few-shot samples. In contrast, our model remains robust and does not experience performance deterioration as the number of training samples increases.

Quantitatively, across the seven datasets, MDGFM outperforms the second-best model by up to 18.04% in the one-shot scenario and 8.11% in the few-shot scenarios.

We further evaluate large $K$ values on the large-scale Penn94 dataset. As shown in Table 3, our model's performance improvement becomes more pronounced as $K$ increases. In contrast, the performance of GCOPE and MDGPT remains relatively stable, showing minimal change.

## 6.3. Ablation Study (RQ2)

In this section, we analyze the effectiveness of key components in our model by designing four variants and comparing

their classification performance against MDGFM.
*w/o-refinedadj*: Removes the GSL module.
*w/o-sumtoken*: Excludes the shared token.
*w/o-topology*: Constructs the graph without considering topological information in the GSL process, i.e., merely use $X_i'$ in Eq.(3) rather than $[X_i', A_i^r X_i']$ for the $i$-th graph.
*w/o-balance*: Removes the balance token.

**Effectiveness of domain-invariant learning.** MDGFM captures shared information across domains through a shared token. As shown in Figure 2, the *w/o-sumtoken* variant exhibits the weakest performance across most datasets, underscoring the importance of capturing common information for effective domain generalization.

**Effectiveness of topology structure alignment.** MDGFM utilizes graph structure learning to extract information from graph topology structure. The *w/o-refinedadj*, *w/o-topology*, *w/o-balance* generally perform much worse than MDGFM, demonstrating the necessity of each component.

## 6.4. Domain Sensitivity (RQ3)

In this section, we examine the extent to which our model depends on specific source domains and its performance under limited source-domain conditions. To assess this, we systematically remove subsets of source domains and evaluate the model using Cornell as the target domain, comparing its performance against multi-domain pre-training baselines. Specifically, we exclude one to three source domains while keeping the target domain and all other parameters fixed.

According to the one-shot result in Table 4, the performance of GCOPE and MDGPT deteriorates significantly, whereas MDGFM maintains strong performance and stability despite the removal of source domains. Our findings suggest that these methods extract varying degrees of knowledge from specific source domains. A counterintuitive phenomenon is that in some cases, removing some source domains leads to

*Table 2.* Cross-domain transfer learning performance (mean accuracy ± std) of few-shot node classification, where the values within parentheses represent the $K$ values used in the context of $K$-shot learning.

| Methods | Cora(5) | Citeseer(5) | Pubmed(5) | Cornell(3) | Squirrel(3) | Chameleon(5) | Penn94(5) |
|---|---|---|---|---|---|---|---|
| GCN | 60.15±5.33 | 45.54±4.71 | 57.82±8.26 | 39.53±13.57 | 21.61±4.22 | 22.09±0.99 | 52.07±1.01 |
| GAT | 59.79±3.89 | 50.48±2.94 | 57.55±9.37 | 34.53±13.01 | 20.11±3.11 | 20.83±1.52 | 50.98±1.23 |
| DGI | 56.76±11.29 | 42.67±8.98 | 54.04±11.59 | 43.22±5.84 | 20.23±1.12 | 27.68±5.21 | 50.13±0.81 |
| GraphCL | 61.59±5.71 | 47.05±6.85 | 58.50±7.38 | 32.77±6.23 | 21.18±0.96 | 27.45±2.58 | 51.84±1.58 |
| GPPT | 43.67±7.11 | 47.31±6.93 | 40.47±10.17 | 34.69±8.54 | 22.14±1.53 | 28.25±1.39 | 51.75±1.60 |
| GPF | 51.21±11.44 | 56.90±8.84 | 58.76±7.70 | 38.17±8.15 | 21.62±3.10 | 28.09±4.93 | OOM |
| GCOPE | 54.63±3.98 | 53.18±4.47 | 57.74±2.73 | 48.21±11.97 | 21.37±4.20 | 25.50±1.23 | 50.69±0.81 |
| MDGPT | 59.64±5.73 | 52.71±5.71 | 58.65±7.54 | 35.18±8.90 | 21.42±4.16 | 26.18±5.18 | 50.41±3.13 |
| MDGFM | **64.56±7.29** | **61.24±4.82** | **63.50±5.81** | **49.56±6.92** | **23.00±4.39** | **30.54±2.87** | **53.58±0.83** |

(a) Adding edges (Cornell)  (b) Deleting edges (Cornell)  (c) Meta-attack (Cornell)  (d) Meta-attack (Cora)

*Figure 3.* Performance of robustness analysis, where the dataset in parentheses represents the target domain.

*Table 3.* $K$-shot node classification accuracy on Penn94.

| $K$ | GCOPE | MDGPT | MDGFM |
|---|---|---|---|
| 10 | 50.30±1.03 | 50.59±4.35 | **54.43±0.93** |
| 50 | 51.36±0.28 | 50.73±3.85 | **55.69±1.28** |
| 100 | 51.61±0.33 | 51.14±4.88 | **58.51±1.15** |
| 500 | 52.88±0.40 | 51.52±3.64 | **63.25±0.82** |

*Table 4.* Performance of domain sensitivity analysis.

| Removed Data | GCOPE | SAMGPT | MDGFM |
|---|---|---|---|
| -Pubmed | 33.38±8.19 | 31.42±9.53 | 39.66±5.51 |
| -Squirrel | 28.88±8.57 | 33.02±8.26 | 40.78±5.44 |
| -Cora-Citeseer | 30.25±8.99 | 30.04±8.77 | 39.96±5.99 |
| -Pubmed-Squirrel | 30.50±8.41 | 31.10±7.43 | 42.68±5.64 |
| -Citeseer-Cora-Pubmed | 26.37±10.85 | 29.38±9.39 | 40.08±5.74 |
| -Pubmed-Citeseer-Squirrel | 31.50±13.18 | 30.90±7.61 | 42.20±5.02 |
| Original Performance | 34.02±11.94 | 33.58±10.38 | 40.77±5.96 |

improved performance. This occurs because multi-domain graphs may result in conflicts or interferences rather than synergy (Yu et al., 2025). Our model aims to mitigate such negative influences by extracting the shared informative and beneficial aspects of the available knowledge.

## 6.5. Robustness Analysis (RQ4)

To evaluate the robustness of our proposed MDGFM, we assess the node classification performance under diverse attacks. Simple modification attacks are implemented by randomly adding or removing edges. Considering the varying influence of individual domains on the target domain, we apply these random attacks to all source domains and the target domain under one-shot learning.

Furthermore, we employ Metattack (Zügner & Günnemann, 2019), an advanced adversarial attack that perturbs the train-

ing data to degrade the model performance after training. Under identical attack conditions, we compare MDGFM against multi-domain methods. As shown in Figure 3, the performance of GCOPE and MDGPT deteriorates as attack intensity increases. Notably, our model is quite stable and consistently outperforms all baselines across all attack scenarios, which can be attributed to our proposed topology alignment. Further analysis, including a sensitivity study of hyperparameters, is provided in Appendix D. Time complexity analysis is also summarized in Appendix B to show the efficiency of MDGFM.

# 7. Conclusion

In this paper, we introduce MDGFM, a novel graph foundation model that unifies graphs from diverse domains into a universal semantic space. Our approach enhances domain generalization by extracting domain-invariant knowledge through both feature and topology alignment. We validate the effectiveness of MDGFM through theoretical analysis and empirical evaluation. To the best of our knowledge, this is the first work to explicitly address invariant information embedded in topology structures across homophilic and heterophilic domains. Our findings pave the way for advancements in graph-based domain generalization, with potential extensions to dynamic graphs and large-scale heterogeneous networks, further enhancing adaptability in real-world applications.

# Impact Statement

This paper aims to advance the field of Machine Learning. While our work has potential societal implications, we do not identify any specific concerns that require particular emphasis at this stage.

# Acknowledgments

This work was supported by the National Natural Science Foundation of China (No. 62276053).

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

## A. Notations

Table 5. Frequently used notations.

| Notation | Description |
|---|---|
| $G_i = (A_i, X_i^{ori})$ | The $i$-th original input graph. |
| $G_{i1} = (A_i, X_i')$ | Graph $i$ after feature projection. |
| $G_{i2} = (A_i', X_i')$ | Graph $i$ after feature projection topology aware alignment. |
| $T = (A_T, X_T)$ | The downstream target graph. |
| $X_i^{ori}$ | Features of the original input graph. |
| $X_i$ | Graph features after dimension alignment. |
| $X_i'$ | Features after feature projection. |
| $Y$ | The set of all possible class labels. |
| $r$ | The order of graph aggregation. |
| $d'$ | Dimension of the original node features. |
| $d$ | The unified dimension. |
| $V$ | The set of nodes. |
| $E$ | The set of edges. |
| $Z$ | Node embeddings. |
| $D$ | Set of source domains. |
| $D_i$ | The $i$-th domain from the source domain set. |
| $D_T$ | Target domain that does not belong to the source domain set. |
| $Proj(\cdot)$ | A specific projection operation, here we use PCA. |
| $\sigma(\cdot)$ | Non-linearity activation function. |
| $\mathcal{L}$ | Loss function. |
| $\mathcal{L}_{dst}$ | Downstream loss function. |
| $sim(\cdot, \cdot)$ | Cosine similarity. |
| $\odot(\cdot)$ | Element-wise multiplication. |
| $t_{D_i}$ | Domain token for domain $D_i$. |
| $t_S$ | Shared token. |
| $t_{B_i}$ | Balance token for domain $D_i$. |
| $p_m$ | Meta prompt. |
| $p_s$ | Specific prompt. |
| $\alpha_i$ | Trainable coefficients for source domain $i$. |
| $\beta$ | A learnable parameter used for integrating prompts. |

## B. Time complexity analysis

In the **pre-training phase**, each of the $N$ source graphs is processed independently. For a graph with $|V|$ nodes and $|E|$ edges (here we select $|V| = max\{V_n\}, |E| = max\{E_m\}, n, m \in [1, N]$), the model first aligns node features via truncated PCA, which reduces the input dimension from $d'$ to $d$ at a cost of $\mathcal{O}(|V| \cdot d \cdot \log d')$. As for token lightweight element-wise multiplication, the time complexity is $|V|d$. It then applies locality-sensitive hashing $k$NN for graph structure learning. Denote the batch size of sparse $k$NN as $B$, each requiring $\mathcal{O}(d)$ operations, resulting in $\mathcal{O}(|V| \cdot B \cdot d)$ time for reconstructing the graph. Next, an $L$-layer GCN operates on the refined structure , contributing an additional $\mathcal{O}(L \cdot |V| \cdot d^2 + L \cdot |E| \cdot d + |V|d)$. Therefore, the total pre-training complexity across all source graphs is $\mathcal{O}\left(N \cdot \left[|V| \cdot d \cdot \log d' + |V| \cdot B \cdot d + L \cdot |V| \cdot d^2 + L \cdot |E| \cdot d\right]\right)$.

Similarly, in the **downstream phase**, PCA procedure takes $\mathcal{O}(|V_T| \cdot d \cdot \log d')$ time. Prompt fusion and token modulation cost $\mathcal{O}(|V_T| \cdot d)$. GSL again uses local sensitive $k$NN, which adds $\mathcal{O}(|V_T| \cdot B \cdot d)$. The GCN encoder then performs $L$-layer message passing with cost $\mathcal{O}(L \cdot |V_T| \cdot d^2 + L \cdot |E_T| \cdot d + |V_T|d)$. Finally, classification is done via prototype matching, where each node compares with $C$ class centroids, yielding $\mathcal{O}(|V_T| \cdot C \cdot d)$. Summing these terms, the overall downstream complexity is $\mathcal{O}(|V_T| \cdot d \cdot \log d' + |V_T| \cdot B \cdot d + L \cdot |E_T| \cdot d + L \cdot |V_T| \cdot d^2 + |V_T| \cdot C \cdot d)$.

Overall, the model scales linearly with the number of nodes and edges, and benefits from efficient structure refinement and modular design like local sensitive $k$NN. We will include this complexity analysis in the final version for completeness.

## C. Datasets

Table 6. Statistics of Datasets.

| Datasets | Nodes | Edges | Feature dimension | Node classes | Homophily ratio |
|---|---|---|---|---|---|
| Cora | 2,708 | 10,556 | 1,433 | 7 | 0.810 |
| Citeseer | 3,327 | 9,104 | 3,703 | 6 | 0.736 |
| Pubmed | 19,717 | 88,648 | 500 | 3 | 0.802 |
| Squirrel | 5,201 | 396,846 | 2,089 | 5 | 0.222 |
| Chameleon | 2,277 | 62,792 | 2,325 | 5 | 0.231 |
| Cornell | 183 | 298 | 1,703 | 5 | 0.305 |
| Penn94 | 41,554 | 2,724,458 | 4,814 | 2 | 0.510 |

We provide an in-depth explanation of datasets, where their statistics details are presented in Table 6.

**Citation network**: The Cora and Citeseer (Sen et al., 2008) datasets represent a varied collection of scholarly articles in the field of computer science, where each node is associated with bag-of-words features and a categorical label denoting the corresponding topic of the paper. In contrast, the Pubmed (Namata et al., 2012) dataset consists of articles focused on diabetes sourced from the PubMed database. Each node in this dataset is characterized by an attribute vector, along with a label that identifies the specific type of diabetes addressed in the publication.

**WebKB**: Cornell (Pei et al., 2020) is a subset derived from the WebKB dataset. In this dataset, each node corresponds to a web page, while the edges illustrate the hyperlinks that connect these pages. The features of nodes are expressed using bag-of-words features derived from the content of the web pages. Nodes are classified into five distinct labels: student, project, course, staff, and faculty.

**Wikipedia network**: The Chameleon and Squirrel (Pei et al., 2020) datasets comprise two page-to-page networks sourced from Wikipedia, each centered on specific themes. Nodes signify individual web pages, and edges represent their connections. Node attributes are characterized by collections of nouns gathered from the content of the pages. Additionally, each node is categorized according to the average monthly traffic that the corresponding web page receives.

**Online social network**: The Penn94 (Traud et al., 2012) dataset is a large-scale social network derived from the Facebook 100 networks of university students from 2005, where each node corresponds to an individual student. The nodes are annotated with the reported gender of each user, and the objective is to predict this gender. The features associated with the nodes include major, secondary major/minor, dorm/house, year, and high school (Lim et al., 2021). The homophily ratio of the Penn94 dataset is 0.51, indicating that its graph topology exhibits a mixture of homophilic and heterophilic patterns. Previous research has shown that such topological information is nearly meaningless, making it difficult for existing topology-fixed GNNs to adapt effectively to this type of graph (Chien et al., 2021; Chen et al., 2024).

## D. Additional Experiments

### D.1. Expansion of Robustness Experiments

We perform robustness experiments on the Cornell dataset before. To investigate whether the MDGFM model retains its robustness under the condition where the downstream target domain consists of homophilic graphs, we replicate the robustness experiments previously described on the Cora dataset, performance under metaattack is shown in the main text. Specifically, we apply random edge additions and deletions on all domains. We compare our results with those of MDGPT and GCOPE, as summarized in Figure 4. Our observations indicate that our model maintains commendable robustness when evaluated on homophilic graph datasets. Furthermore, we note that in the context of heterophilic graph datasets, GCOPE exhibits greater robustness compared to MDGPT, while the opposite trend is observed for homophilic graphs.

### D.2. Sensitivity Analysis

We investigate the impact of the hyperparameter $k$ in $k$-nearest neighbors ($k$NN) for downstream graph structure learning, which is crucial to our model. The performance of our model varies with changes in the hyperparameter $k$ as illustrated in Figure 5. Overall, our model exhibits low sensitivity to changes in $k$, regardless of whether the downstream target domain is

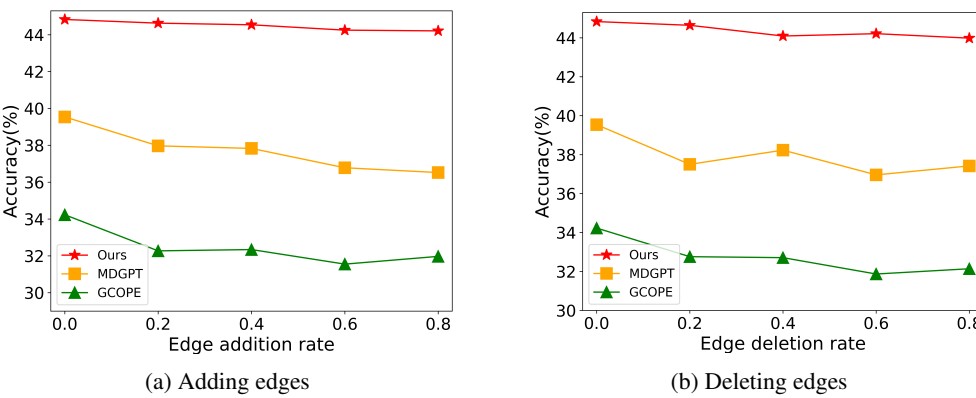

*Figure 4.* Performance of robustness analysis, where Cora is the target domain.

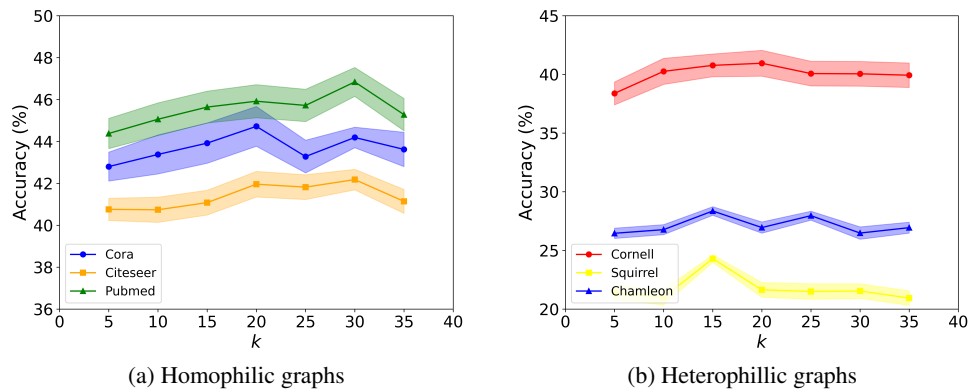

*Figure 5.* Sensitivity study of $k$.

homophilic or heterophilic. When the target domain is homophilic graphs, larger values of $k$ (such as 20 or 30) yield the best results. In contrast, for heterophilic graphs, relatively smaller values of $k$ can produce satisfactory outcomes.

## E. Experimental Details

For both one-shot and few-shot classification tasks, we pretrain the models on five datasets and subsequently perform predictions on the remaining dataset, ensuring that the downstream domain remains unseen during the training phase. The detailed experimental setup is summarized in Table 7, where a checkmark ($\checkmark$) indicates visibility during pre-training, while the absence of the mark denotes invisibility. For the large-scale dataset Penn94, it is exclusively used as the target domain to evaluate the generalization capability of the model.

All experiments are conducted on a platform equipped with an Intel(R) Xeon(R) Gold 5220 CPU and an NVIDIA A800 80GB GPU, using PyTorch 1.10.1 and DGL 0.9.1. Each experiment is run five times and the average results are reported. We employ the Adam optimizer and set the batch size to 128. During the upstream pre-training phase, we utilize Principal Component Analysis (PCA) to reduce the dimensionality of the initial features to 50 dimensions, thereby unifying the features from multiple source domains. For homogeneous graphs, we set $k$=30 for Graph Structure Learning (GSL), while for heterophilic graphs, we configure $k$=15. Additionally, we adjust the upstream and downstream learning rates across different datasets, as detailed in Table 8. We fix the number of graph neural layers to 3, with a hidden dimension of 256 for the GCN model. When dealing with homophilic graphs, the number of Epochs is set to 60, whereas it is set to 100 for heterophilic graphs. For the downstream node classification tasks, we implement few-shot learning scenarios with 1-shot and 5-shot settings (3-shot for the Cornell and Squirrel datasets). We perform 50 resampling iterations in downstream few-shot classification. The $k$ values required for graph construction adhere to the configurations established during the upstream pre-training phase.

*Table 7.* Settings of multi-domain transfer.

| Target domain | Cora | Citeseer | Pubmed | Squirrel | Chameleon | Cornell |
|---|---|---|---|---|---|---|
| Cora |  | ✓ | ✓ | ✓ | ✓ | ✓ |
| Citeseer | ✓ |  | ✓ | ✓ | ✓ | ✓ |
| Pubmed | ✓ | ✓ |  | ✓ | ✓ | ✓ |
| Squirrel | ✓ | ✓ | ✓ |  | ✓ | ✓ |
| Chameleon | ✓ | ✓ | ✓ | ✓ |  | ✓ |
| Cornell | ✓ | ✓ | ✓ | ✓ | ✓ |  |
| Penn94 | ✓ | ✓ | ✓ | ✓ | ✓ | ✓ |

*Table 8.* The hyperparameters corresponding to each dataset.

| Target domain | Pre-training learning rate | Downstream learning rate | Epoch | Unified dimension | Dropout | Downstream $k$ |
|---|---|---|---|---|---|---|
| Cora | 0.0075 | 0.001 | 60 | 50 | 0.1 | 30 |
| Citeseer | 0.001 | 0.001 | 60 | 50 | 0.1 | 30 |
| Pubmed | 0.0001 | 0.0015 | 60 | 50 | 0.1 | 30 |
| Squirrel | 0.01 | 0.0003 | 100 | 50 | 0.1 | 15 |
| Chameleon | 0.02 | 0.01 | 100 | 50 | 0.1 | 15 |
| Cornell | 0.02 | 0.0003 | 100 | 50 | 0.1 | 15 |
| Penn94 | 0.0001 | 0.003 | 100 | 50 | 0.1 | 30 |

# F. Methodology Details

Following the approach proposed in previous works (Liu et al., 2022; Shen et al., 2024), after constructing the cosine similarity matrix of $H_i$, we implement post-processing techniques to ensure that $A_i'$ exhibits the characteristics of sparsity, non-negativity, symmetry, and normalization. For the convenience of discussion, the subscript $i$ is omitted hereinafter.

$k$**NN for sparsity**. In most applications, a fully connected adjacency matrix often has limited practical significance and incurs high computational costs. Therefore, we employ the $k$-Nearest Neighbors ($k$NN) operation to sparsify the learned graph. For each node, we retain the edges with the top-$k$ values and set the others to 0, thereby obtaining the sparse adjacency matrix $A^{sp}$. Note that we employ efficient $k$NN with locality-sensitive hashing (Fatemi et al., 2021) to enhance the model's scalability. This approach avoids the resource-intensive computation and storage of explicit similarity matrices, reducing the complexity from $\mathcal{O}(|V|^2)$ to $\mathcal{O}(|V|B)$, where $|V|$ is the number of nodes and $B$ is the batch size of the sparse $k$NN.

**Symmetrization and Activation**. Since real-world connections are typically bidirectional, we symmetrize the adjacency matrix. Additionally, the weight of each edge should be non-negative. Given the input $A^{sp}$, these operations can be expressed as follows:

$$A^{sym} = \frac{\sigma(A^{sp}) + \sigma(A^{sp})^\top}{2} \tag{10}$$

where $\sigma(\cdot)$ represents a non-linear activation implemented by the ReLU function.

**Normalization**. The normalized adjacency matrix with self-loops can be obtained as follows:

$$A' = (\tilde{D}^{sym})^{-\frac{1}{2}} \tilde{A}^{sym} (\tilde{D}^{sym})^{-\frac{1}{2}} \tag{11}$$

where $\tilde{D}^{sym}$ is the degree matrix of $\tilde{A}^{sym}$ with self-loops. Subsequently, for each view, we can acquire the adjacency matrix $A_i'$, which possesses the desirable properties of sparsity, non-negativity, symmetry, and normalization.

# G. Details of Loss Functions

Note that directly computing mutual information between two graphs is impractical due to the complexity of graph-structured data. Since we focus on node-level tasks, we assume the optimized graph should guarantee that each node's neighborhood substructure contains sufficient task-relevant information. Therefore, this requirement can be transferred to mutual information between node representations (Liu et al., 2024), which can be easily computed using a sample-based

differentiable lower bound. For any view $i$ and $j$, the lower bound $I_{lb}$ of the mutual information $I(Z^i; Z^j)$ is (Liang et al., 2023):

$$I_{lb}(Z^i; Z^j) = \mathbb{E}_{\substack{z^i, z^{j+} \sim p(z^i, z^j) \\ z^j \sim p(z^j)}} \left[ log \frac{expf(z^i, z^{j+})}{\sum_N expf(z^i, z^j)} \right] \tag{12}$$

where $f(\cdot, \cdot)$ is a score critic approximated by a neural network. $p(z^i, z^j)$ denotes the joint distribution of node representations from views $i$ and $j$, while $p(z^i)$ denotes the marginal distribution. $z^i$ and $z^{j+}$ are mutually positive samples, representing the representations of the same node in views $i$ and $j$ respectively.

Note that in our work, we redefine $I(G_{i1}; G_{i2} \dagger A_i')$, where $A_i'$ is the refined adjacency matrix to identify the positive samples. That is to say, $z^i$ and $z^{j+}$ are mutually positive samples if and only if node $i$ and node $j+$ are neighbors in matrix $A_i'$. Recall the following loss function in the pre-training phase:

$$\mathcal{L} = -I(G_{i1}; G_{i2} \dagger I_e) - I(G_{i1}; G_{i2} \dagger A_i') \tag{13}$$

where $I_e$ is the identity matrix. Particularly, the second term incorporates the refined graph structure $A_i'$ to increase the number of positive samples. The minimization objective $\mathcal{L}$ in multi-domain pre-training is calculated as follows:

$$\ell_{I_e}(Z_m^{i1}, Z_m^{i2}) = log \frac{e^{sim(\tilde{Z}_m^{i1}, \tilde{Z}_m^{i2})/\tau_c}}{\sum_{n=1}^{|V|} e^{sim(\tilde{Z}_m^{i1}, \tilde{Z}_n^{i2})/\tau_c}}, \quad \ell_{A_i'}(Z_m^{i1}, Z_m^{i2}) = log \frac{\sum_{n=1, A_i'[m,n] \neq 0}^{|V|} A_i'[m, n] \cdot e^{sim(\tilde{Z}_m^{i1}, \tilde{Z}_m^{i2})/\tau_c}}{\sum_{n=1}^{|V|} e^{sim(\tilde{Z}_m^{i1}, \tilde{Z}_n^{i2})/\tau_c}} \tag{14}$$

$$\hat{\mathcal{L}} = -\frac{1}{2} \sum_{m=1}^{|V|} (\ell_{I_e}(Z_m^{i1}, Z_m^{i2}) + \ell_{I_e}(Z_m^{i2}, Z_m^{i1})) - \frac{1}{2} \sum_{m=1}^{|V|} (\ell_{A_i'}(Z_m^{i1}, Z_m^{i2}) + \ell_{A_i'}(Z_m^{i2}, Z_m^{i1})) \tag{15}$$

where $Z^{i1}$ denotes the node representations of the original graph $G_{i1} = (A_i, X_i')$ obtained through the graph encoder, and $Z^{i2}$ denotes the node representations of the refined graph $G_{i2} = (A_i', X_i')$. $\tilde{Z}_m^{i1}$ is the non-linear projection of $Z_m^{i1}$. $A_i'[m, n]$ denotes the element in the $m$-th row and $n$-th column of the adjacency matrix $A_i'$, which corresponds to the edge weights of the learned structure via GSL. $sim(\cdot, \cdot)$ denotes the cosine similarity. $\tau_c$ is the temperature parameter, which is fixed at $0.2$ in this work. During the pre-training phase, we sequentially minimize the objective $\hat{\mathcal{L}}$ for each input graph with different domains.

