# OpenReview forum: "Multi-Domain Graph Foundation Models: Robust Knowledge Transfer via Topology Alignment"
_ICML.cc/2025/Conference — ICML 2025 poster_

### Official Review · Reviewer_QrBx · 2025-03-06

**Overall Recommendation:** 4

**Summary:**

This paper proposes the Multi-Domain Graph Foundation Model (MDGFM) to address the challenge of transferring knowledge across graphs from different domains. MDGFM aligns graph topologies through a decoupled embedding mechanism, a graph structure learning module, and a prompt-tuning approach. This alignment allows MDGFM to effectively transfer knowledge from multiple source domains to a target domain, even for unseen domains. Theoretical analyses and experiments on both homophilic and heterophilic graph datasets validate the robustness and efficacy of MDGFM.

**Claims And Evidence:**

Yes

**Essential References Not Discussed:**

There is a new publication, which is highly related to this submission. The authors should take a look.
SAMGPT: Text-free Graph Foundation Model for Multi-domain Pre-training and Cross-domain Adaptation, WWW 2025.

**Experimental Designs Or Analyses:**

Yes, the experiments are sound to me.

**Methods And Evaluation Criteria:**

Yes, the methodology and evaluation make sense to me.

**Other Comments Or Suggestions:**

The font for k is different in the caption and Fig.5.

**Other Strengths And Weaknesses:**

Strengths
The paper is well-structured and easy to follow, with a clear presentation of the methodology and results.
The authors provide a solid theoretical foundation for MDGFM, including proofs of its effectiveness and domain generalization capabilities.
The proposed MDGFM shows robust performance across various datasets, including both homophilic and heterophilic graphs.
Weaknesses
The authors should provide more detailed explanations of the experimental results, particularly why certain methods outperform others in specific scenarios.
The authors should include more large-scale datasets to validate the scalability of the proposed method.

**Questions For Authors:**

Could you provide more insights into why MDGFM performs better on certain homophilic and heterophilic datasets?
How does MDGFM handle imbalanced datasets or noisy data?
Have you considered applying MDGFM to dynamic graphs or temporal datasets?

**Relation To Broader Scientific Literature:**

Existing graph models, such as graph neural networks (GNNs), heavily depend on labeled data, which is often scarce and costly. This paper proposes an effective and robust unified graph foundation model, which performs well on graphs in different domains.

**Theoretical Claims:**

Yes, I checked them.

---

> ### Author Rebuttal · Authors · 2025-03-31
>
> We appreciate your thoughtful feedback. Your constructive criticism is invaluable in refining our work. Below, we give point-by-point responses to your comments.
>
> **Weakness 1 & Question 1: Further explanations**
>
> Thank you for raising this important point. We agree that a clearer explanation of MDGFM’s performance across different datasets enhances the completeness of our work. To address it comprehensively, we now analyze the performance from two perspectives: task setting (one-shot vs. multi-shot transfer) and graph homophily (homophilic vs. heterophilic).
>
> (i) One-shot vs. Multi-shot
>
> **In one-shot settings**, the target domain has few labeled data, so adaptation heavily relies on domain alignment and structural generalization. Thus, methods like GCOPE and MDGPT which pretrain from multiple domains get better results than supervised or graph prompting methods. Specifically, MDGFM outperforms all baselines due to its knowledge transferring capacity. It effectively captures both domain-specific information and shared patterns across domains.
> **In multi-shot settings**, baselines like GraphCL and GPF (even GCN) occasionally catch up or outperform multi-domain pre-training methods. This is because they leverage target supervision more directly, and in high-label regimes, their implicit overfitting can yield short-term gain. However, MDGFM remains competitive and often more stable under cross-domain generalization due to invariant learning.
>
> (ii) Homophilic vs. Heterophilic Graphs
>
> **On homophilic graphs**, traditional GNNs like GCN or GAT may perform decently due to strong neighborhood-label consistency. However, MDGFM still shows advantages in low-label settings due to cross-domain transferability via decoupled embedding and prompt regularization.
>
> **On heterophilic graphs, MDGFM significantly outperforms almost all baselines.** This is because standard message passing methods suffer from abundant noise resided in topology structure, while our framework applies GSL to learn invariant knowledge robustly, learning domain-aligned graphs that reduce harmful interference.
>
> **Weakness 2: Additional experiments on large-scale datasets**
>
> We sincerely appreciate your suggestion to include large-scale evaluations. In response, we carefully selected and added **three additional large-scale datasets** with 30K+ nodes (i.e., Github, Deezer and T-Finance), covering diverse graph domains and scales, to thoroughly validate the scalability and generalization ability of MDGFM. Experimental settings are same as previous setups. New results along with the original Penn94 evaluation are summarized in **https://anonymous.4open.science/r/Large-scale-datasets-35EE**. It shows that MDGFM still outperform all baseline methods and shows great scalability.
>
> **Question 2: Additional experiments on imbalanced datasets and noisy data**
>
> Thanks for your concern. We would like to clarify that our paper includes robustness analysis under multi-type noisy conditions in Section 6.5. Expansion of robustness analysis could be seen in Appendix C.1. Specifically, we simulate noise by randomly adding or deleting edges, even conducting meta-attack. Results demonstrate that MDGFM consistently outperforms baselines under all scenarios, which confirms its robustness to structural noise.
>
> In response to the reviewer's suggestion, we conduct **new experiments on imbalanced data** using a real-world financial graph dataset T-Finance, which exhibits high class imbalance with skewed label distributions (minority-majority ratio up to 0.048:1). We compare MDGFM with GCOPE and MDGPT under same training setups using ACC/F1/AUC metrics.
>
> The results, summarized in **https://anonymous.4open.science/r/Imbalanced-Noisy-data-05ED**, show that MDGFM achieves significantly better performance confirming its effectiveness under imbalanced label regimes. We believe these results, together with the original noisy-graph experiments, provide a comprehensive view of its robustness.
>
> **Question 3: Generalization to Dynamic or Temporal Graphs**
>
> Currently, MDGFM is designed for static graphs. However, due to its modular and decoupled structure, it can be extended to dynamic settings in future work. Specifically the prompt and structure learning modules can be adapted to handle temporal snapshots. We appreciate this insightful suggestion and will discuss temporal extensions in the Conclusion section.
>
> Additionally, thank you for pointing out the relevant work SAMGPT(WWW 2025), which is the published version of citation [Yu et al., 2024]. We also acknowledge the difference in the font of "k", and will correct it in the final version.
>
> Once again, we thank the reviewer for the constructive suggestions. With the added experiments, domain analyses, and scalability discussion, we believe MDGFM has strong theoretical grounding, broad practical utility, and extensibility toward future challenges. We hope these improvements merit a stronger overall recommendation.

---

> > ### Comment · Reviewer_QrBx · 2025-04-09
> >
> > Thank you for your detailed response—it has resolved my concerns. I would like to recommend the acceptance of this paper.

---

### Official Review · Reviewer_6nDL · 2025-03-07

**Overall Recommendation:** 4

**Summary:**

The authors propose MDGFM to solve the graph pre-training issue. The key contributions include: A novel framework that aligns graph topologies across multiple domains using Graph Structure Learning (GSL);An adaptive embedding mechanism that balances features and topologies for improved generalization; A dual-prompt tuning approach that enhances adaptation to unseen domains. Extensive experiments validate the model’s effectiveness.

**Claims And Evidence:**

Yes

**Essential References Not Discussed:**

The problem of multi-domain generalization in graph learning is important, and the paper presents a timely solution inspired by the success of foundation models in NLP and CV.

**Experimental Designs Or Analyses:**

Yes

**Methods And Evaluation Criteria:**

Yes

**Other Comments Or Suggestions:**

See Strengths And Weaknesses.

**Other Strengths And Weaknesses:**

Strengths:

The problem of domain generalization in graph learning is essential, and the paper presents a timely solution inspired by the success of foundation models in NLP and CV. The topology alignment mechanism and graph structure refinement are well-motivated, addressing key challenges in cross-domain graph learning. The paper evaluates adversarial robustness and domain sensitivity, showing the model’s resilience to noise and distribution shifts.

Weaknesses: The following aspects could further strengthen it.

1. The computation complexity is unclear. Given the increasing size of real-world graphs, a complexity analysis would improve clarity.

2. The paper emphasizes the role of meta-prompts and specific prompts, but it does not extensively analyze their individual contributions.

**Questions For Authors:**

Q1: How are the hyperparameters selected, such as the dimensionality of the feature space and the number of neighbors in kNN?

Q2: How does MDGFM perform with and without prompt tuning?

**Relation To Broader Scientific Literature:**

The problem of multi-domain generalization in graph learning is important, and the paper presents a timely solution inspired by the success of foundation models in NLP and CV.

**Theoretical Claims:**

Yes

---

> ### Author Rebuttal · Authors · 2025-03-31
>
> We sincerely thank the reviewer for the positive and constructive feedback. We greatly appreciate your recognition of the importance of the problem, the motivation of our proposed components, and the comprehensive empirical evaluation. Below we address your valuable suggestions and questions.
>
> **Weakness 1: Computational complexity**
>
> Thanks for your concern. While our method introduces additional modules such as GSL and prompt-tuning, we have carefully designed MDGFM to remain computationally efficient and scalable to real-world graphs. Following your advice, we provide a detailed analysis of the computational complexity of MDGFM for both the pre-training and downstream phases. In the **pre-training phase**, each of the $N$ source graphs is processed independently. For a graph with $|V|$ nodes and $|E|$ edges (here we select $|V|=max\\{V_n\\},|E|=max\\{E_m\\},n,m \in [1,N]$), the model first aligns node features via truncated PCA, which reduces the input dimension from $d'$ to $d$ at a cost of $\mathcal{O}(|V| \cdot d \cdot \log d')$. As for token lightweight element-wise multiplication, the time complexity is $|V|d$. It then applies locality-sensitive hashing $k$NN for graph structure learning. Denote the batch size of sparse $k$NN as $B$, each requiring $\mathcal{O}(d)$ operations, resulting in $\mathcal{O}(|V| \cdot B \cdot d)$ time for reconstructing the graph. Next, an $L$-layer GCN operates on the refined structure , contributing an additional $\mathcal{O}(L\cdot |V|\cdot d^2+L \cdot |E| \cdot d+|V|d)$. Therefore, the total pre-training complexity across all source graphs is $\mathcal{O}\left(N \cdot \left[ |V| \cdot d \cdot \log d' + |V| \cdot B \cdot d +L\cdot |V|\cdot d^2+ L \cdot |E| \cdot d \right]\right)$.
>
> Similarly, in the **downstream phase**, PCA procedure takes $\mathcal{O}(|V_T|\cdot d \cdot log d^{\prime})$ time. Prompt fusion and token modulation cost $\mathcal{O}(|V_T| \cdot d)$. GSL again uses local sensitive $k$NN, which adds $\mathcal{O}(|V_T| \cdot B \cdot d)$. The GCN encoder then performs $L$-layer message passing with cost $\mathcal{O}(L\cdot |V_T|\cdot d^2+L \cdot |E_T| \cdot d+|V_T|d)$. Finally, classification is done via prototype matching, where each node compares with $C$ class centroids, yielding $\mathcal{O}(|V_T| \cdot C \cdot d)$. Summing these terms, the overall downstream complexity is $\mathcal{O}(|V_T| \cdot d \cdot \log d'+|V_T| \cdot B \cdot d + L \cdot |E_T| \cdot d +L\cdot |V_T|\cdot d^2+ |V_T| \cdot C \cdot d)$.
>
> Overall, the model scales linearly with the number of nodes and edges, and benefits from efficient structure refinement and modular design like local sensitive $k$NN. We will include this complexity analysis in the final version for completeness.
>
> **Weakness 2 & Question 2**
>
> Thank you for highlighting this important point. Following your suggestions, we conduct **new ablation studies** and include the results in **https://anonymous.4open.science/r/Ablation-study-on-prompts-FD62**. Specifically, we compare four variants of the model: Full MDGFM (with both meta- and specific prompts), w/o meta-prompt (only using target-specific prompt), w/o specific prompt (only using global meta-prompt), and w/o both prompts (i.e., no prompt tuning at all). The results show that:
>
> 1. Removing the specific prompt leads to a noticeable performance drop, especially in domains with strong local structural patterns, confirming its role in target adaptation. Removing the meta-prompt also hurts performance, particularly in low-shot settings, indicating that it captures generalizable cross-domain knowledge. Note that it is normal that meta-prompt causes relatively slight impact than specific prompt.
> 2. Obviously, removing both results in degradation, confirming that the two prompts are complementary and essential for effective transfer.
>
> These findings support our design choice of dual-prompt tuning and clarify their respective impacts.
>
> **Question 1: Hyperparameters**
>
> We appreciate the reviewer’s attention to experimental details. For feature projection, we apply PCA to reduce the dimensionality of all node features to **$d = 50$**. This value is chosen based on empirical studies and balances expressiveness with computational efficiency. For $k$NN graph construction, we use different $k$ values depending on the structural characteristics of the graph: For homophilic graphs we use $k = 30$ to keep more original relations between nodes. As for heterophilic graphs, we use a smaller $k = 15$ to avoid amplifying noisy connections. We have also provided a sensitivity analysis (Appendix C), which shows that MDGFM remains robust across a range of $k$ values.
>
> To further demonstrate the robustness of our model to different hyperparameter choices, we additionally conduct a **sensitivity analysis** on the feature dimension $d$, and compare the results against GCOPE and MDGPT. The experimental results are summarized in **https://anonymous.4open.science/r/Sensitivity-on-d-6510**.

---

### Official Review · Reviewer_XcGn · 2025-03-09

**Overall Recommendation:** 4

**Summary:**

The authors propose a unified approach that aligns graph topologies and features across domains, leveraging Graph Structure Learning (GSL) to refine noisy and adversarial-prone real-world graphs. The framework also introduces an efficient prompt-tuning mechanism to enhance knowledge transfer to unseen domains.

**Claims And Evidence:**

Well supported.

**Essential References Not Discussed:**

Quite complete.

**Experimental Designs Or Analyses:**

It is convincing.

**Methods And Evaluation Criteria:**

It is convincing.

**Other Comments Or Suggestions:**

1."facilitate robust knowledge transfer." → Consider "facilitate effective and robust knowledge transfer." for clarity.
2.The error bound theorem is strong, but consider discussing potential limitations or assumptions

**Other Strengths And Weaknesses:**

1. The paper is well-structured and presents a valuable contribution. It addresses a critical gap in the field of graph representation learning by focusing on topology alignment across diverse domains.
2. The introduction of domain tokens and shared tokens for semantic alignment is innovative and effectively bridges the gap between domains with varying structural and feature characteristics.
3. The experimental evaluation is comprehensive. The results demonstrate consistent improvements over state-of-the-art baselines in both one-shot and few-shot learning scenarios.
4. Some intuitive explanations could be given. For example, the paper introduces several new components (e.g., domain tokens, shared tokens, balance tokens, dual prompts) that may make it not accessible to a broader audience.

**Questions For Authors:**

Refer to the weaknesses.

**Relation To Broader Scientific Literature:**

Current graph models often struggle with generalization due to challenges such as graph heterogeneity and scarcity of domain-specific data. Creating robust and adaptable graph foundation models is the next big thing for practical applications.

**Theoretical Claims:**

Correct.

---

> ### Author Rebuttal · Authors · 2025-03-31
>
> We sincerely thank the reviewer for the highly encouraging and constructive feedback. We are especially grateful for your recognition of our model’s generalization capability, methodological contributions, and comprehensive evaluations. Below we address your valuable suggestions.
>
> **Weakness: Intuitive explanations of model components**
>
> We sincerely thank the reviewer for pointing out the potential accessibility issue due to the introduction of multiple novel components.  We agree that intuitive explanations will improve the clarity of our method, especially for broader audiences not specialized in cross-domain graph learning.  Below, we provide a brief intuitive summary of each key component:
>
> **Domain Tokens** ($t_{D_i}$): Each domain token acts like a value vector in a Transformer model, storing domain-specific knowledge during pretraining. During downstream phase, the target domain serves as a query to selectively retrieve and apply relevant knowledge from the source domains.  This design enables flexible and efficient cross-domain transfer through implicit attention-like behavior.
>
> **Shared Token** ($t_S$): This acts like a "semantic anchor" shared across all domains. It helps extract and preserve invariant patterns that reside in multiple domains, enabling better cross-domain alignment.
>
> **Balance Token ($t_{B_i}$)**: This component adaptively balances the contribution of node features and graph topology. Intuitively, it acts as a "tuner" that decides how much structural information to retain versus how much feature content to emphasize, especially helpful when structural noise or heterophily is present.
>
> **Dual Prompts** (meta-prompt and specific-prompt): These serve as "adapters" during downstream transfer.    Meta-prompt transfers generalized knowledge learned from source domains, while the specific prompt fine-tunes the model to the unique structure and features of the target domain.
>
> In the revised version, we will incorporate these intuitive explanations into the methodology section (Sec. 4) to improve accessibility without sacrificing technical depth.
>
> **Comment 1: "facilitate robust knowledge transfer." → Consider "facilitate effective and robust knowledge transfer." for clarity.**
>
> We appreciate the suggestion. We will revise the corresponding phrase in the refined version.
>
> **Comment 2: Limitations and assumptions of error bound theorem**
>
> Thank you for highlighting this. Our theoretical results (Theorems 5.1 and 5.3) rely on the covariate shift assumption and existence of invariant subgraphs across domains. In the revised paper, we will explicitly enumerate the following limitations:
>
> 1. The error bound assumes that the target distribution lies within (or close to) the convex hull of the source domains. In highly diverse or outlier domains, this assumption may be violated.
> 2. One potential limitation of our theoretical framework lies in the assumption of the existence of a universal invariant graph learner $\Phi^*$. This assumption requires that core semantic and structural patterns are preserved across domains after graph structure learning (GSL). However, in real-world scenarios where the relationship between topology and features varies significantly across domains—e.g., in one domain, structure dominates label prediction, while in another, node features are more informative—such shared invariances may not naturally exist. In these cases, identifying a single $\Phi^*$ that captures consistent and transferable structural knowledge across all domains becomes highly non-trivial. We acknowledge this as a theoretical boundary condition and note that our empirical results suggest MDGFM remains effective even when this assumption is only approximately satisfied.
>
> Despite these theoretical assumptions, we observe in our ablation and robustness experiments (Sections 6.3–6.5) that MDGFM maintains strong performance even under domain removal and adversarial perturbation, which empirically validates the practical soundness of the theoretical setup.
>
> Once again, we sincerely appreciate your time and effort in reviewing our paper. Your constructive criticism has been invaluable in refining our work, and we are more than happy to add clarifications to address any additional recommendations and reviews from you!

---

### Decision · Program_Chairs · 2025-05-01

**Decision:**

Accept (poster)

**Comment:**

All reviewers acknowledged the effectiveness and novelty of the proposed method of graph foundation models and recommended acceptance. The rebuttals addressed some minor issues related to the computational complexity of the proposed algorithm and experiment on larger datasets. In sum, this is a high-quality work and should be accepted.